# Development of a Lightweight Pavement Block with Extremely High Permeability Using the Volcanic Pumice Bora

**Kentaro Yasui [1]**, **Yuri Sakaida [1]**, **Kenshiro Yamamura [2]**, **Makoto Minamimagari [3]**, **Eitaro Horisawa [4]**, **Chihiro Morita [4]** and **Hiroyuki Kinoshita [4],***

[1]  National Institute of Technology, Kagoshima College, 1460-1 Shinko, Hayato-cho, Kirishima 899-5193, Japan; yasui@kagoshima-ct.ac.jp (K.Y.); y.sakaida.aa@c-nexco-het.jp (Y.S.)
[2]  Graduate School of Engineering, University of Miyazaki, 1-1 Gakuen-Kibanadai-Nishi, Miyazaki 889-2192, Japan; kensirou3@icloud.com
[3]  Nanken Kogyo Co., Ltd., 5629-2 Yamada-Karuishi, Yamada-cho, Miyakonojo 889-4601, Japan; contact@nanken-kogyo.com
[4]  Department of Engineering, University of Miyazaki, 1-1 Gakuen-Kibanadai-Nishi, Miyazaki 889-2192, Japan; horisawa.eitaro@cc.miyazaki-u.ac.jp (E.H.); cgmorita@cc.miyazaki-u.ac.jp (C.M.)
*  Correspondence: t0d165u@cc.miyazaki-u.ac.jp; Tel.: +81-985-58-7290

**Abstract:** Bora is a granular volcanic pumice with a certain degree of hardness. In this study, we investigated the possibility of using fired Bora as a substitute for fine sand in mortar. The objective was to develop a lightweight pavement block with extremely high permeability using fired Bora. Initially, the changes in hardness and density of Bora particles fired at different temperatures were examined. The bending strength of mortar containing fired Bora as a fine aggregate was also evaluated. Subsequently, we fabricated a permeable pavement block with a hybrid structure, comprising a mortar containing the fired Bora and a porous material with large-sized Bora particles bonded using cement paste. We examined its bending strength and permeability and performed a quantitative analysis of the bending stress distribution on the block using the finite element method. The hardness of Bora significantly increased when fired at temperatures exceeding 900 °C; hence, Bora fired at 1100 °C can be used as an effective substitute for crushed sand in mortar. Furthermore, the results confirmed that the use of fired Bora enabled the production of lightweight pavement blocks with extremely high permeability that satisfied the bending strength criterion.

**Keywords:** volcanic pumice; water permeable pavement block; bending strength; permeability; finite element analysis

## 1. Introduction

A large area of the Southern Kyushu region of Japan is occupied by several volcanoes, calderas, and the "Shirasu Plateau", formed by volcanic eruption deposits. This plateau in Kagoshima Prefecture, Japan, accounts for more than 50% of the total area. The strata contain significant volumes of Shirasu soil [1], a glassy volcanic material with fine particle size, and a volcanic pumice referred to as "Bora" [2], which particle size is larger than that of Shirasu. Shirasu and Bora are classified as special soils [3] by the Japanese Ministry of Agriculture, Forestry, and Fisheries, because the Shirasu Plateaus are susceptible to erosion by rainwater and groundwater; moreover, landslides occur often during heavy rainfall [4,5]. The soil is unsuitable for rice cultivation owing to its poor water retention capacity and lack of nutrients necessary for rice plants [6].

Although volcanic soil and pumice can be effectively utilized, they have limited applications. In agriculture, potatoes and soybeans are planted to utilize Shirasu soil properties. In industries, Shirasu is used to produce porous glass. However, no reports exist on applications using the volcanic pumice Bora. Bora exhibits a certain degree of water absorption and retention owing to its porosity. Although Bora is occasionally used

as a soil conditioner, its use in agriculture is limited. Moreover, as the hardness of Bora is lower than that of common gravel and sand, it is rarely used in civil engineering or as a construction material, such as a concrete aggregate [7–9].

In recent years, abnormal weather conditions induced by global warming have caused localized heavy rainfall exceeding 100 mm/h, resulting in flood damage [10,11]. Therefore, the urgent requirement for countermeasures has led to the installation of water-permeable pavement blocks on the sidewalks in urban areas to reduce flood damage. However, the lack of sand resources and obtaining high hardness sand as aggregates in mortar and concrete has become a global problem [12,13] and has resulted in the development of various artificial aggregates [14–16].

In this study, we examined the possibility of using fired Bora as a substitute for crushed sand in mortar. Furthermore, we developed a pavement block with extremely high permeability and low weight using fired Bora. The changes in hardness and density at different temperatures were initially examined, and the strength of mortar containing fired Bora as a fine aggregate was evaluated. Subsequently, we developed a permeable pavement block with a hybrid structure using mortar containing the fired Bora and a porous material that comprised large-sized fired Bora particles bonded with cement paste. Finally, the bending strength and permeabilities of the permeable blocks were examined.

Additionally, the pavement block was subjected to a bending stress analysis using the finite element method (FEM) to determine the internal stress state when a bending moment is applied to the permeable pavement block. This aided in determining the appropriate dimensions of the permeable section. The results of the FEM analysis were cross-referenced with the bending strength values of each material that constituted the permeable block, and the dimensions of the permeable section that satisfied the bending strength criteria for permeable pavement blocks were identified. Furthermore, we analyzed the permeability of the blocks, which depended on the dimensions of the permeable section.

## 2. Materials and Methods

### 2.1. Fundamental Chemical and Physical Properties of Bora

Figure 1 depicts microscopic images of Bora (Nanken Kogyo Co., Ltd., Miyakonojo, Japan), fired Bora, and crushed sand (Gaiatec Co., Ltd., Sendai, Japan). Bora was mined from the area surrounding Mount Kirishima in Southern Kyushu, Japan. The particle size of Bora was adjusted to 2 mm or less by sieving. The fired Bora was heated at 100 °C/h to a firing temperature of 1100 °C in an electric furnace (KY-4N, Kyoei Electric Kilns Co., Ltd., Tajimi, Japan). The samples were maintained at the firing temperature for 1 h and allowed to cool to approximately 25 °C in the furnace. Typically, crushed sand is used as a fine aggregate in mortar and concrete. The strength of the mortar using Bora as the fine aggregate was compared to that of the mortar using crushed sand; the particle size of the sand was also less than 2 mm. Here, the Bora is fired to increase the strength (hardness) of the Bora itself.

Table 1 lists the chemical compositions of Bora and Shirasu, which were used as reference materials. An energy-dispersive X-ray analyzer (EDX-720; Shimadzu Corporation, Kyoto, Japan) was used for these measurements. Bora and Shirasu exhibited similar compositions, and their primary components were silica and alumina. However, the calcium content in Bora was slightly higher than that in Shirasu.

Table 2 lists the densities and water absorptions of the samples measured according to the Japanese industrial standard JIS A 1109: 2020 [17], "Methods of test for density and water absorption of fine aggregates". The density of Bora was approximately three-fourths that of crushed sand, and the water absorption was substantially higher than that of the crushed sand; this implies that Bora comprised numerous pores in its structure.

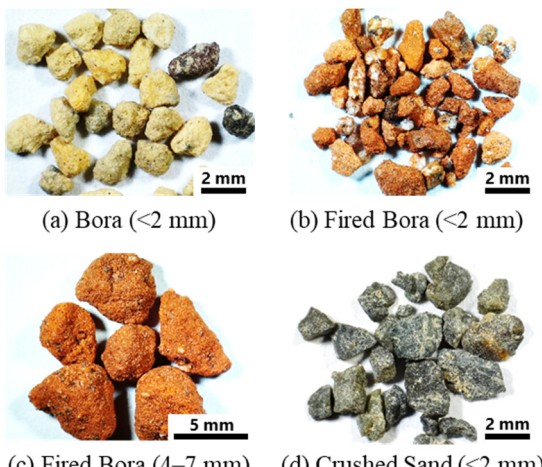

**Figure 1.** Microscopic images of un-fired and fired Bora samples and crushed sand.

**Table 1.** Chemical compositions of the raw materials.

| Component | Volcanic Soils | |
|---|---|---|
| | Bora (%) | Shirasu (%) |
| $SiO_2$ | 64.9 | 66.2 |
| $Al_2O_3$ | 22.6 | 25.2 |
| $Fe_2O_3$ | 4.89 | 2.62 |
| $K_2O$ | 2.41 | 2.55 |
| MgO | - | 1.68 |
| CaO | 4.10 | 1.36 |
| $TiO_2$ | 0.56 | 0.26 |
| Others | 0.54 | 0.13 |

**Table 2.** Densities and water absorptions of fine aggregates.

| Fine Aggregate | Density (g/cm$^3$) | | Water Absorption (%) |
|---|---|---|---|
| | Saturated and Surface-Dry | Oven-Dry | |
| Un-fired Bora | 1.91 | 1.58 | 20.8 |
| Fired Bora | 2.26 | 2.08 | 9.45 |
| Crushed sand | 2.85 | 2.80 | 1.61 |

*2.2. Manufacturing of the Permeable Pavement Block*

Figure 2 illustrates a schematic of a permeable pavement block developed from fired Bora. The outer frame of the block was composed of mortar material containing fired Bora particles with sizes of 2 mm or less. Its interior was composed of a porous material with fired Bora of sizes ranging from 4 to 7 mm bonded with cement paste.

The permeable pavement block was assembled by placing a compact porous material at the center of the product formwork. Mortar was filled around the porous material and was allowed to flow into the formwork by vibratory compaction. After demolding, the sample was cured in water for 28 d at 20 ± 1 °C.

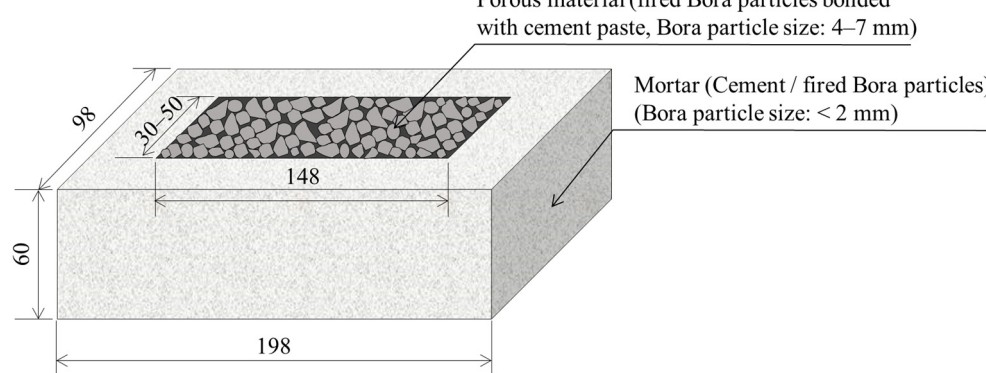

**Figure 2.** Schematic of a permeable pavement block (all units in mm).

Table 3 lists the mix design of the mortar material. Initially, the Bora particles or sand, ordinary Portland cement (Taiheiyo Cement Corporation, Bunkyo-ku, Japan), and water were mixed in a mortar mixer (MIC-362-1-01, MARUI & Co., Ltd., Daito, Japan). Subsequently, rectangular blocks with dimensions of 40 mm × 40 mm × 160 mm were prepared separately for the bending tests according to the JIS R 5201:2015 [18], "Physical testing methods for cement".

**Table 3.** Mix designs of mortar materials.

| Type of Mortar | Water/Cement (W/C) | Unit Quantity (g/L) | | |
| --- | --- | --- | --- | --- |
| | | Water | Cement | Aggregate (Particle Size: <2 mm) |
| Un-fired Bora/cement | | | | 1557 |
| Fired Bora/cement | 0.5 | 260 | 520 | 1120 |
| Sand/cement | | | | 1048 |

Table 4 summarizes the mix designs of porous material in which un-fired or fired Bora particles were bonded with cement paste. The porous material contained less water than the mortar material to maintain gaps between the Bora particles. Similarly, rectangular blocks with dimensions of 40 mm × 40 mm × 160 mm were prepared separately for the bending tests.

**Table 4.** Mix designs of porous materials.

| Type of Porous | W/C | Unit Quantity (g/L) | | |
| --- | --- | --- | --- | --- |
| | | Water | Cement | Un-Fired or Fired Bora (Particle Size: 4–7 mm) |
| Bora (un-fired) | 0.23 | 42 | 183 | 576 |
| Fired Bora | | | | 614 |

*2.3. Experimental Methods*

2.3.1. Properties of Bora Particles

The Vickers hardness of un-fired and fired Bora particles at 600–1100 °C was measured using a Vickers hardness testing machine (HMV-2000, Shimadzu Corp., Kyoto, Japan) according to JIS Z 2244:2009 [19], "Vickers hardness test— Part 1: Test method".

The X-ray diffraction (XRD) profiles were obtained using a X-ray crystal structure analyzer (X'Pert-Pro MRD, Malvern Panalytical, Enigma, UK) to examine the changes in the crystal structures of the Bora particles before and after firing. Furthermore, the

specific surface area and pore size distribution of the samples were measured using a high-precision gas/vapor adsorption measurement instrument (BELSORP-max, MicrotracBEL Corp., Osaka, Japan) to examine the changes in the size and number of pores present in the fired Bora.

### 2.3.2. Bending Tests

We performed three-point bending tests to examine the bending strengths of the mortar, porous materials, and permeable pavement blocks. As mentioned in Section 2.2, the dimensions of the mortar and porous materials used for the bending tests were 40 mm × 40 mm × 160 mm. The permeable pavement block used for the bending tests was a sample with the dimensions and shapes illustrated in Figure 2.

The distance between the lower supporting points in the three-point bending tests of mortar and porous materials was 100 mm. The distance between the lower supporting points in the bending test on the permeable pavement block was 160 mm. Five samples of each type of block were used for the analysis. Bending tests on the mortar and porous materials were performed using a precision universal testing machine (AG-X50kN, Shimadzu Corp., Kyoto, Japan) at a crosshead speed of 5 mm/min. The bending strength of the sample was calculated using the maximum load. Similarly, the bending tests on the permeable pavement block were performed using a universal testing machine (MIE-734-1-20, MARUI & Co., Ltd., Daito, Japan) at a crosshead speed of 5 mm/min.

### 2.3.3. Permeability Tests

Figure 3 depicts a schematic of the permeability test performed on the permeable pavement block. The test was performed according to JIS A 5371:2016 [20], "Precast unreinforced concrete products". The block was placed in a steel formwork, and the gap between the formwork and the block was filled using a rubber sheet. Water was poured on the upper surface of the block at approximately 6 L/min, and the amount of water passing through the block per unit time was measured. The coefficient of the block permeability was calculated using Equation (1):

$$k = \frac{Q \times t}{A \times h \times 30(\text{s})} \times 10^{-2}, \tag{1}$$

where $k$ (cm/s) denotes the permeability coefficient, $Q$ (cm$^3$) indicates the amount of water drained in 30 s, $t$ (cm) represents the thickness of the pavement block, $A$ (cm$^2$) denotes the cross-sectional area of the block, and $h$ (cm) indicates the difference in the hydraulic head. The measurements for each of the five blocks used in the test were obtained three times to calculate the average permeability coefficient.

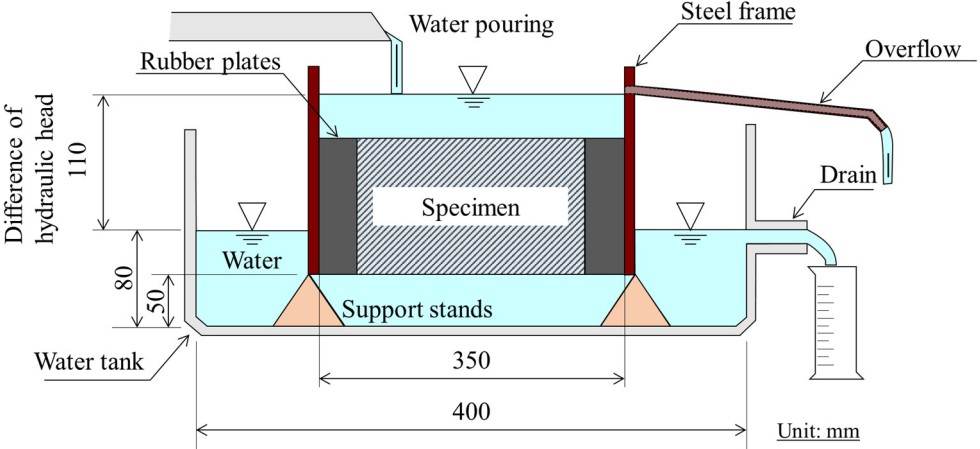

**Figure 3.** Schematic of a permeability test.

### 2.4. Bending Stress Analysis of Permeable Pavement Blocks Using FEM

Figure 4 depicts an example of the computational model used to estimate the internal stress of a permeable pavement block subjected to a bending moment. The peripheral dimensions of the model are 100 mm in width, 60 mm in height, and 200 mm in length, and the distance between the lower supporting points is 160 mm.

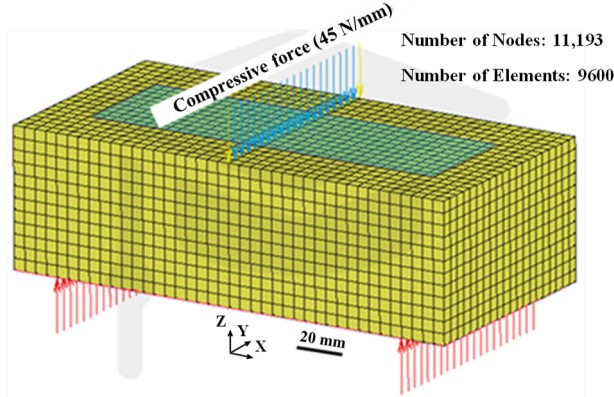

**Figure 4.** Computational model of a permeable pavement block.

We used three types of models for the FEM analysis. Model I was a pavement block entirely composed of mortar. Model II was only an outer frame of the permeable pavement block, composed of a mortar material without a porous material inside the block; the dimensions of the hollow part inside the block were 40 mm × 60 mm × 150 mm. Model III was the permeable pavement block fabricated in this study. The outside of the permeable pavement block was composed of mortar, and the inside of the block was composed of a porous material. The dimensions of the porous material inside the block were 40 mm × 60 mm × 150 mm.

Marc/Mentat 2022 software was used to perform the bending stress analysis of the block. Table 5 lists the computational conditions in the FEM analysis. The densities of the mortar and porous materials were measured according to JIS A 1109:2020 [17], "Methods of test for density and water absorption of fine aggregates". The values of Young's moduli were obtained from compressive tests for each material, and the Poisson's ratio was assumed to be 0.2 [21,22].

**Table 5.** Computational conditions of the finite element method (FEM) analysis.

| Type of Material | Density (kg/m$^3$) | Poisson's Ratio | Young's Modulus (GPa) |
|---|---|---|---|
| Mortar | 2020 | 0.2 | 16.2 |
| Porous | 1087 | 0.2 | 1.66 |

For the boundary conditions of the model, the deformation along the *z*-direction was constrained at the lower supporting points in all the models. Additionally, the deformation along the *x*-direction of the nodal points in the rectangular cross-section at the middle of the block span was constrained.

A total vertical load of 4500 N (for example, a compressive load of 45 N/mm in Model I or III) was applied to the middle of the upper surface of the block in all the models, and a linear elastic analysis was performed. The load of 4500 N resulted in a bending stress of 3 MPa at the center of the conventional pavement block composed of a homogeneous material, thereby rendering it equivalent to the 3 MPa bending strength criterion required for pavement blocks.

## 3. Results and Discussion

### 3.1. Hardness and Density of Fired Bora

Figure 5a,b illustrate the Vickers hardness values and densities of the fired Bora particles, respectively. Bora particles fired at 900 °C or higher significantly increase in hardness and density. Figure 6 depicts the XRD profiles of the un-fired and fired Bora particles at 1100 °C. The un-fired Bora particles contained paracelsian ($Ba(Al_2Si_2O_8)$) [23] and anorthite ($CaAl_2Si_2O_8$) [24] as the primary minerals. In contrast, the fired Bora particles contained larger amounts of anorthite and hematite ($Fe_2O_3$) [25]. The color of Bora changed to reddish-brown after firing owing to the crystallization of the iron component (Figure 1).

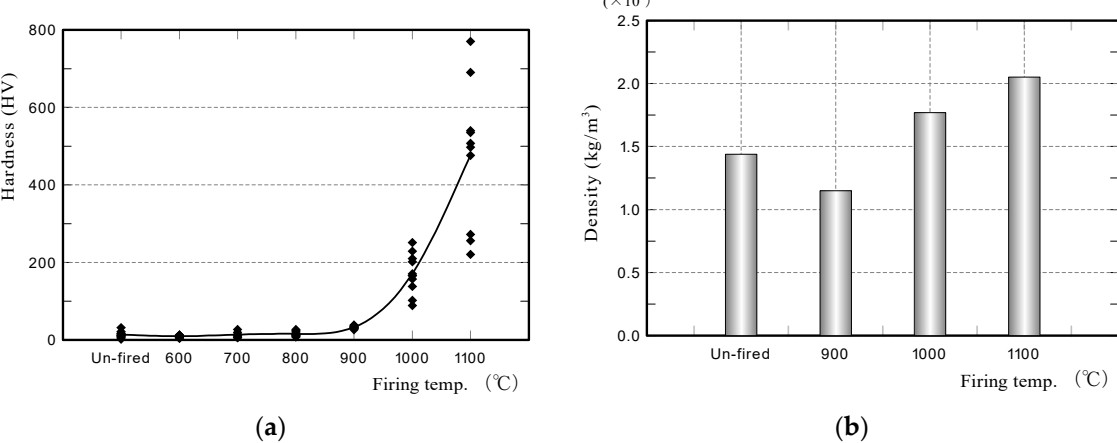

**(a)**  **(b)**

**Figure 5.** (**a**) Vickers hardness test results, and (**b**) densities of the fired Bora particles.

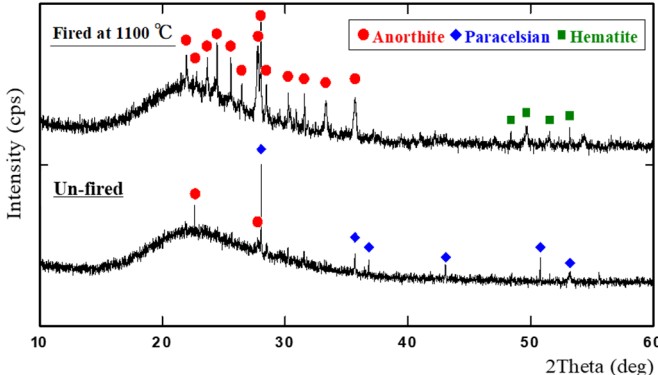

**Figure 6.** X-ray diffraction (XRD) profiles of Bora particles.

Figure 7 illustrates the relationship between the pore size distribution and firing temperature of the Bora particles. The number of nanometer-sized pores decreased with the increased firing temperature, thereby decreasing their specific surface areas. In particular, pores of sizes 100 nm or less in the sample fired at 1100 °C were not detected. This indicated that the fine pores in the structure disappeared owing to the sintering and densification of the structure.

The factors affecting the hardening of the fired Bora particles were investigated based on the aforementioned measurement results. No significant difference existed between the hardness of paracelsian (6 on the Mohs scale) and that of anorthite (6–6½ on the Mohs scale) and hematite (5–6 on the Mohs scale). Therefore, the hardness of the minerals in the sample was not likely to increase.

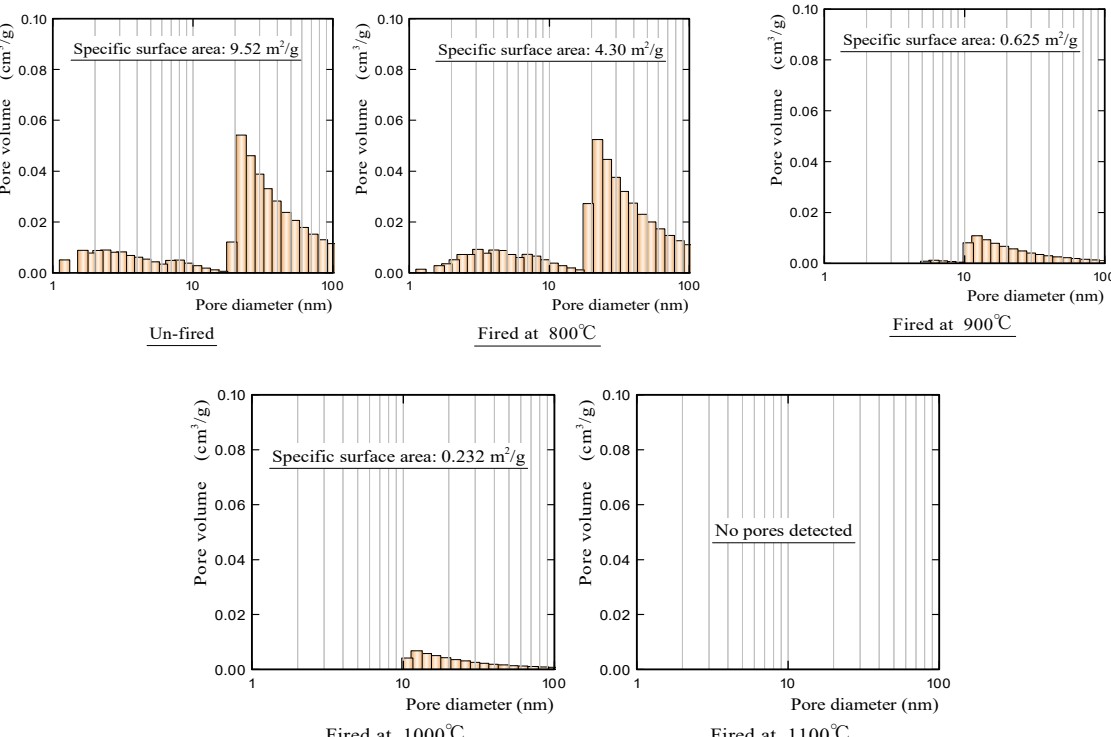

**Figure 7.** Pore size distributions of un-fired and fired Bora particles.

Anorthite is a Ca-rich plagioclase. In the active CaO-SiO$_2$-Al$_2$O$_3$ amorphous materials, the sintering and densification of the structure are initiated at approximately 900 °C, and the material forms polycrystalline sintered compacts when the temperature exceeds approximately 1000 °C [26]. Therefore, the increase in the hardness of fired Bora particles can be primarily attributed to the enlarged crystalline structure caused by the chemical bonding among CaO, SiO$_2$, and Al$_2$O$_3$, as well as the densification of the structure induced by sintering. The enlarged crystalline structure and the densification eliminated the fine pores in the structure, increasing the hardness of the sample. The XRD profiles of the fired Bora particles suggest that the CaO component contained in Bora significantly impacts the hardness of the sample after firing.

### 3.2. Bending Strength of Mortar and Porous Materials Using Fired Bora Particles

Figure 8a illustrates the relationship between the bending strength of mortar materials manufactured under the conditions listed in Table 3 and curing time. Figure 8b depicts the microscopic images of the fractured surfaces after the bending test of the mortar specimens. The bending strength of all the mortar materials increased rapidly, up to approximately 7 d from the start of curing in water, and then increased gradually and became nearly constant after approximately 28 d.

We then considered the effect of the type of fine aggregate on the bending strength of the mortar material. Although the bending strength of a fired Bora/cement mortar material was higher than that of an un-fired Bora/cement mortar material, it was lower than that of the sand/cement mortar material. However, the fired Bora/cement mortar material exhibited considerably higher strength than the bending strength criterion ($\geq$3 MPa) for permeable pavement blocks. Furthermore, the compressive strength of the fired Bora/cement mortar material obtained from a separate compressive test was 33.6 MPa, which satisfies the strength criteria of concrete with lightweight aggregates (JIS A 5002: 2003 [27], JIS A 5308: 2019 [28], and Article 74, Order for Enforcement of the Japanese Building Standards Act [29]). Therefore, the fired Bora/cement mortar material can be a suitable material for constructing permeable pavement blocks. The low bending strength of the Bora/cement

mortar material can be attributed to the hardness being lower than that of sand; the hardness values of the sand, un-fired Bora particle, and fired Bora particle were 512, 14, and 476 HV, respectively. As the hardness of the un-fired Bora particle was considerably lower than that of the sand, it cannot be a suitable material for permeable pavement blocks.

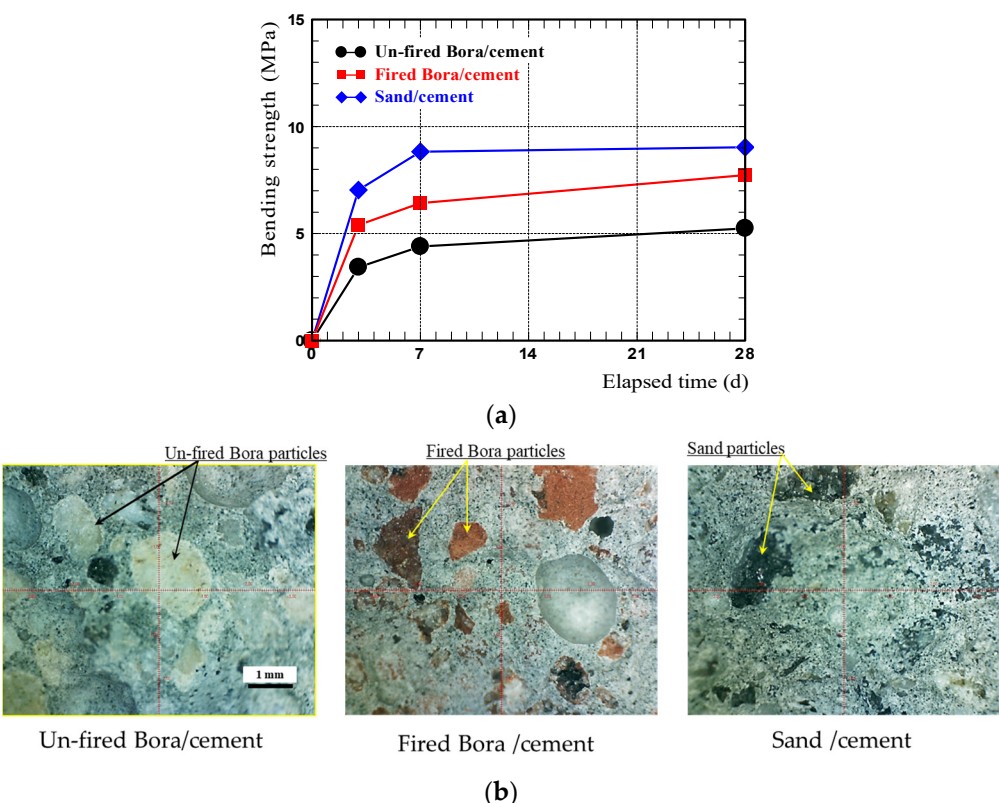

**(a)**

**(b)**

**Figure 8.** (**a**) Bending strength of the mortar materials, and (**b**) microscopic images of their fractured surfaces after the bending test.

As indicated in Figure 8b, the differences in the hardness of the fine aggregate contained in the mortar material resulted in differences in the fracture morphology of the mortar specimens. In other words, the fired Bora/cement mortar and sand/cement mortar were primarily fractured by cracking at the interface between the cement and the Bora particles or sand, whereas the un-fired Bora/cement mortar was fractured owing to the propagation of cracks into Bora particles. Mortar materials with an intragranular fracture are undesirable in the aggregate, because such aggregates cannot strengthen the base material (cement) from the viewpoint of the composite law of strength [30], although the shrinkage of the cement paste could be suppressed to some extent when hardened. Therefore, the un-fired Bora/cement mortar material cannot be utilized, whereas the fired Bora/cement mortar material can be used as a lightweight mortar material.

Figure 9 depicts the bending strengths of the porous materials produced by combining un-fired or fired Bora particles with cement paste. As the strength of the porous materials depended on the hardness of the Bora particles, the porous material with fired Bora particles exhibited a bending strength nearly twice that of the porous material with un-fired Bora particles.

However, the bending strength of the porous material using fired Bora particles was approximately 2.3 MPa, which was lower than the bending strength criterion of 3 MPa for permeable pavement blocks. Therefore, permeable pavement block prototypes should be subjected to bending strength tests and stress analysis using FEM to determine whether the porous material with fired Bora particles can be used to fabricate permeable pavement blocks.

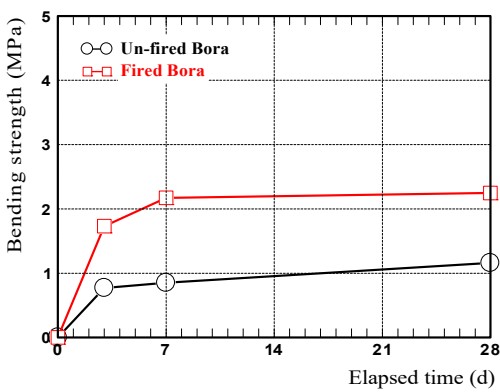

**Figure 9.** Bending strengths of porous materials.

### 3.3. Bending Strength and Permeability of Permeable Pavement Blocks

Figure 10a illustrates the bending strength and water permeability of the permeable pavement block prototypes, and Figure 10b shows the images of the fractured surfaces of the blocks. Two prototypes of permeable pavement blocks were fabricated, one using un-fired Bora particles as the porous material and the other using Bora particles fired at 1100 °C as the porous material. In the case of the mortar material constituting the outer frame of the block, Bora particles fired at 1100 °C were used in both prototypes.

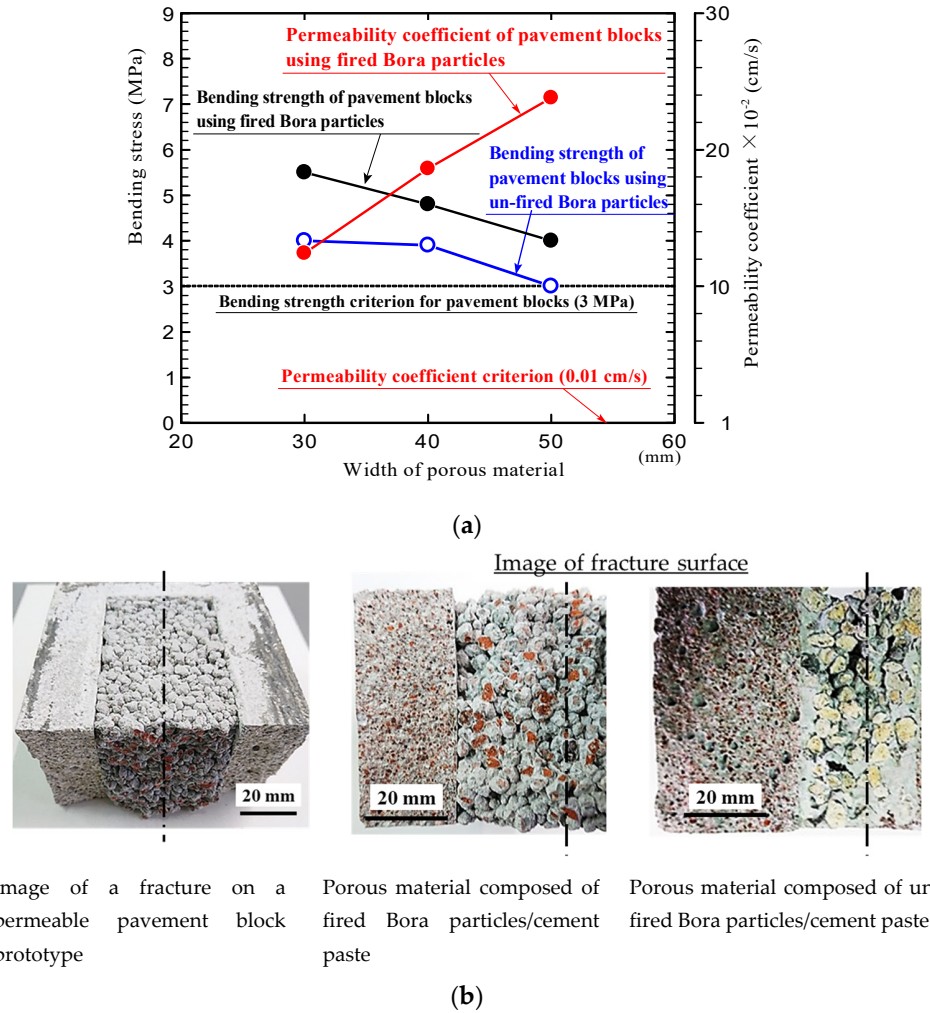

**Figure 10.** (**a**) The bending strength and water permeability of the permeable pavement block prototypes. (**b**) Image of the fractured surfaces of the blocks.

The horizontal axis of the graph represents the width of the porous material section in the pavement block (Figure 2). The bending strength tests were performed on samples with porous material sections that were 30, 40, and 50 mm wide. The bending strength points plotted in the graphs represent the average values of five samples; only the permeability coefficients of pavement blocks composed of fired Bora particles are shown.

The results of the bending tests in Figure 10a confirm that the bending strengths of the permeable pavement blocks using fired Bora as the porous material were relatively higher than those of the blocks using un-fired Bora. The bending strength of the permeable pavement blocks decreased with the increase in the width of the porous material section, because the strength of the porous material was lower than that of the mortar material.

As indicated in Figure 10b, the entire block failed simultaneously with the fracture of the outer frame (mortar material) in all the prototypes. Although no failure was observed in the porous material before the failure of the outer frame of the block, the same differences described in Section 3.2 were observed in the fractured surface of the porous material between the blocks using fired and un-fired Bora particles. In other words, numerous intragranular fractures of un-fired Bora particles were observed in the block that used un-fired Bora particles as the porous material.

The bending strengths of the permeable pavement block with fired Bora particles as the porous material were 4.0–5.5 MPa when the widths of the porous material section were 30–50 mm. Therefore, the blocks satisfied the bending strength criterion for permeable pavement blocks. In contrast, the blocks with un-fired Bora particles satisfied the criterion only when the widths of the porous material sections were within 40 mm; however, the margin against the bending strength criterion was extremely small.

The results of the bending tests of the permeable pavement block prototypes verify that the blocks fabricated using fired Bora particles sufficiently satisfied the strength criterion, despite the bending strength of the porous material being lower than the strength criterion (3 MPa). This indicates that the entire block satisfied the criterion because of the arrangement of the mortar material, which has a relatively high strength on the outside of the porous material. The subsequent section details the stress state of the block based on FEM analysis.

The permeability coefficient of the permeable pavement blocks fabricated using fired Bora particles increased as the porous material section was enlarged; the value was 12 to 24 times larger than the criterion ($1.0 \times 10^{-2}$ cm/s) of the permeable pavement blocks. The water permeability of the blocks was at least 200 mm/h over that reported in [31]. This is because the pavement block contained a porous material with multiple gaps, which was composed of fired Bora particles bonded with cement paste, rendering the blocks extremely lightweight. For instance, the density of the permeable pavement block prototype fabricated using fired Bora with a 50-mm-wide porous section was approximately 1.73 g/cm$^3$.

### *3.4. Results of the Bending Stress Analysis of a Permeable Pavement Block Using FEM*
### 3.4.1. Bending Stress Generated Inside the Block

Figure 11 depicts the contour maps of the normal stress $\sigma_{xx}$ along the *x*-direction for each model obtained via FEM analysis. Figure 12 illustrates the distribution of $\sigma_{xx}$ at nodes along line A—A (Figure 11) in the middle at the bottom of the block. Here, Figures 11 and 12 indicate the stress distributions when the width of the porous material section or the hollow section is 40 mm. Only a slight difference exists between the values of $\sigma_{xx}$ and the maximum principal stress at the initial point.

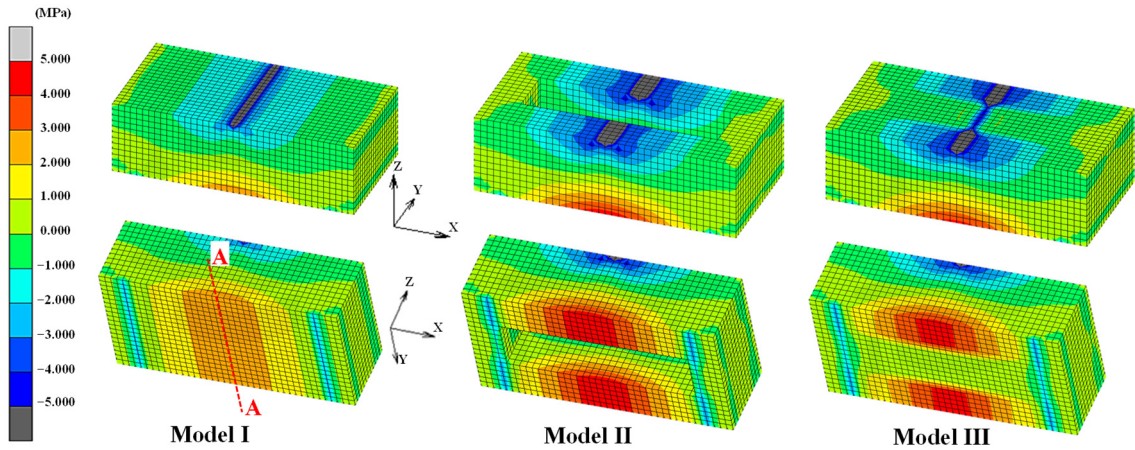

**Figure 11.** Contour maps of the normal stress ($\sigma_{xx}$) along the *x*-direction.

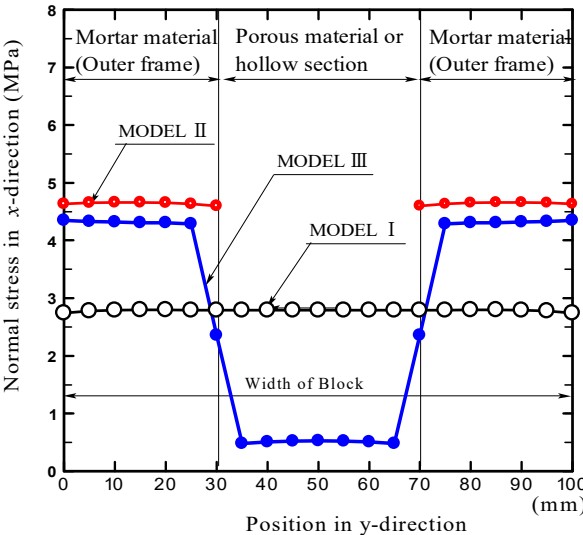

**Figure 12.** Distribution of $\sigma_{xx}$ at nodes along line A—A in the middle at the bottom of the block.

The stress distribution in Model I, wherein the block was composed of mortar alone, was identical to that of the bending stress on a beam supported with a uniform cross-section when a vertical load was applied to the middle of the beam. The maximum tensile stress occurred at the bottom of the rectangular cross-section in the middle of the block length, whereas the maximum compressive stress occurred at the top of the cross-section.

The contour map of Model II indicates the stress distribution when the same load (4500 N) as that of Model I is applied to a hollow block, wherein only the outer frame of the permeable pavement block is modeled. Considering that the stress distribution is the same as that of the stress caused when a vertical load is applied to the middle of a parallel beam with two connected ends, the maximum tensile stress occurs at the bottom of the rectangular cross-section in the middle of the block length on both sides. However, when the same load as Model I was applied to Model II, the internal stress of the hollow block became relatively higher than that of Model I. The value of the maximum principal stress in Model II was approximately 4.7 MPa, which was less than the bending strength of the mortar material (7.5 MPa). Therefore, the hollow block composed only of mortar material can withstand a load of 4500 N, which causes a bending stress of 3 MPa in conventional pavement blocks.

The contour map of Model III illustrates the distribution of normal stress $\sigma_{xx}$ along the *x*-direction on a permeable pavement block composed of mortar and porous materials. Model III exhibits a stress distribution similar to that of Model II; a relatively high stress

occurred in the outer frame composed of mortar, whereas the stress in the porous material was considerably lower. The stress gradient along the *x*-direction in the porous material section was also extremely small.

Figure 12 provides quantitative confirmation that the stress distribution in the outer frame of the block is slightly lower than that in Model II and is higher than that in Model I. Moreover, the stress in the porous material section is substantially lower than that in Model I, and the maximum value of the stress is lower than 3 MPa.

The stress analytical results clarify that, in the permeable pavement block composed of mortar and porous materials, the load is primarily endured by the outer frame (mortal material). Therefore, the permeable pavement block fabricated using fired Bora particles satisfies the bending strength criterion, although the block contains porous material with a bending strength lower than the criterion. This can be attributed to the suppression of the deformation of the porous material inside the block caused by the peripheral mortar material, because the outer frame composed of mortar exhibits a higher rigidity than the porous material. In other words, the stress in the mortar material is higher than that in the porous material when the mortar and porous materials are subjected to the same distortion, because the elastic modulus of the mortar material is higher than that of the porous material.

### 3.4.2. Dimensional Design of the Porous Material Section to Satisfy the Bending Strength Criteria of Permeable Pavement Blocks

Figure 13 depicts the distribution of $\sigma_{xx}$ at nodes along line A—A in the middle of the length of the permeable pavement block at the bottom, obtained via FEM analysis when the widths of the porous material section were changed from 40 mm to 70 mm. Here, the points enclosed by curve A (termed Balloon A) represent $\sigma_{xx}$ induced in the outer frame composed of mortar material, and the points enclosed by curve B (Balloon B) represent $\sigma_{xx}$ generated at the interface between the mortar and porous materials. Similarly, the points enclosed by curve C (Balloon C) represent $\sigma_{xx}$ generated in the porous material.

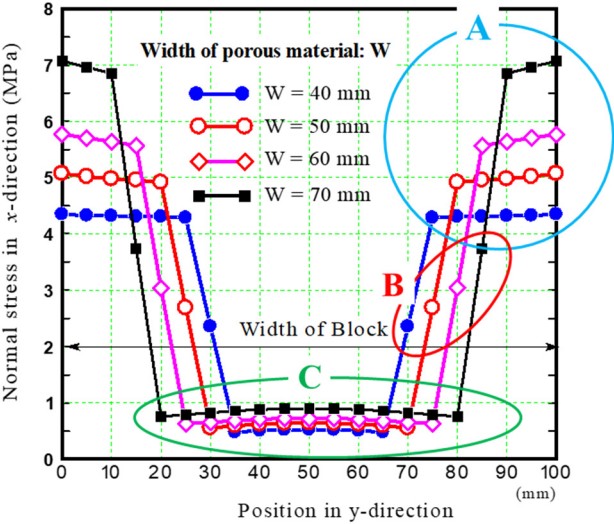

**Figure 13.** Distribution of $\sigma_{xx}$ at nodes along line A—A in the middle of the length of the permeable pavement block at the bottom.

This figure confirms that, as the width of the porous material section increased, the stress in the mortar material increased, whereas the stress in the porous material did not increase significantly.

Figure 14 illustrates the relationship between the width of the porous material section of the permeable pavement block and the strength of each material used to construct the block. The dashed lines in the figure represent the bending strengths of the mortar material, porous material, and cement paste. The bending strength of the mortar and porous

materials was the strength obtained from the bending tests, as described in Section 2.2. The tensile strength of the cement paste was estimated to be approximately 1/13 of the compressive strength of ordinary Portland cement [21,32,33]. The three points plotted along the single dotted line in the figure represent the bending strengths of the permeable pavement block prototypes when the widths of the porous material section are 30, 40, and 50 mm, as discussed in Section 2.4.

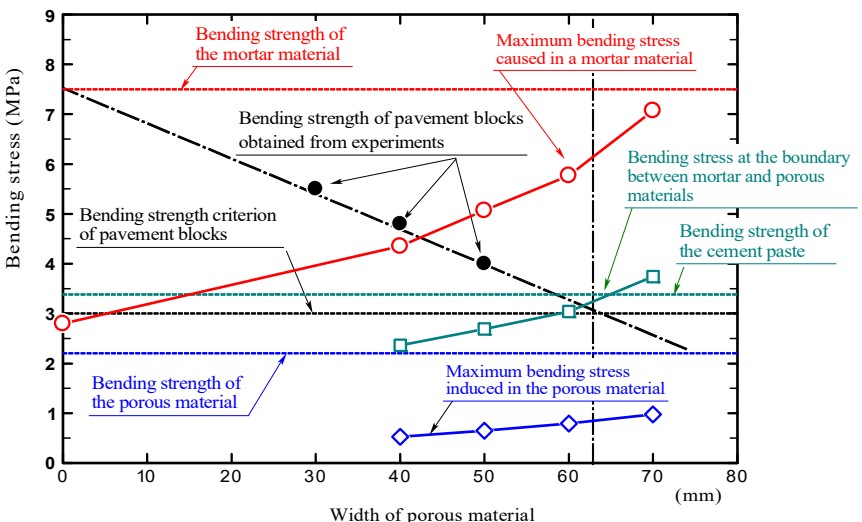

**Figure 14.** Relationship between the width of the porous material of the permeable pavement blocks and the bending strength.

The three curves in the figure indicate the changes caused in the maximum bending stress of the mortar material, porous material, and at the interface between the mortar and porous materials, indicated in the areas of Balloons A, B, and C in Figure 13, respectively, when the widths of the porous material section are changed from 40 mm to 70 mm.

We first discuss the bending strength induced by the mortar material (the outer frame) of the permeable pavement block. The bending strength of the mortar material obtained from the three-point bending test was approximately 7.5 MPa. Conversely, the maximum bending stress generated in the mortar material obtained from the FEM analysis was lower than the bending strength of the mortar material when the width of the porous material section was less than 70 mm; however, the bending stress in the mortar material increased with the increase in the width of the porous material section. Therefore, the outer frame did not fracture.

We then consider the tensile stresses generated at the interface between the mortar and porous materials. The mortar and porous materials are bonded by cement paste. The minimum tensile strength of the cement paste is approximately 3.4 MPa. However, the tensile stress generated at the interface obtained from the FEM analysis was higher than the tensile strength of the cement paste in the porous material section with a width of approximately 60 mm or higher when the width was changed from 40 mm to 70 mm. Therefore, when the width of the porous material section was larger than approximately 60 mm, the mortar and porous materials were separated, eventually causing the porous material to crack or fracture.

In terms of the maximum bending stress generated in the porous material, the stress obtained from the FEM analysis was lower than the bending strength of the porous material when its width ranged between 40 and 70 mm. Therefore, we concluded that the porous material does not fracture unless cracks occur at the interface between the mortar and porous materials.

These results indicate that the mortar material, porous material, and the interface between them do not fail when the porous material width is less than approximately 60 mm. In other

words, the permeable pavement block does not fracture if the width of the porous material section is maintained within 60 mm.

Based on the intersection of the single point and dashed lines representing the bending strength criterion of 3 MPa, the maximum width of the porous material section with respect to the bending strength of the permeable pavement block prototype was approximately 62 mm. The results of the FEM analysis concurred with the experimental results. Both the experimental and analytical results validate that the use of fired Bora particles can produce lightweight pavement blocks with extremely high permeability.

Figure 15a depicts the contour maps of $\sigma_{xx}$ in the permeable pavement block, and Figure 15b illustrates the distribution of $\sigma_{xx}$ at the nodes along line A—A at the bottom of the block, wherein the middle of the upper surface of the block is compressed by a constant strain rate up to $-0.04$ mm in the $z$-direction. The analysis was conducted on the block with a 40-mm-wide porous material. Here, the compressive load was determined based on the reaction force induced at the lower supporting points.

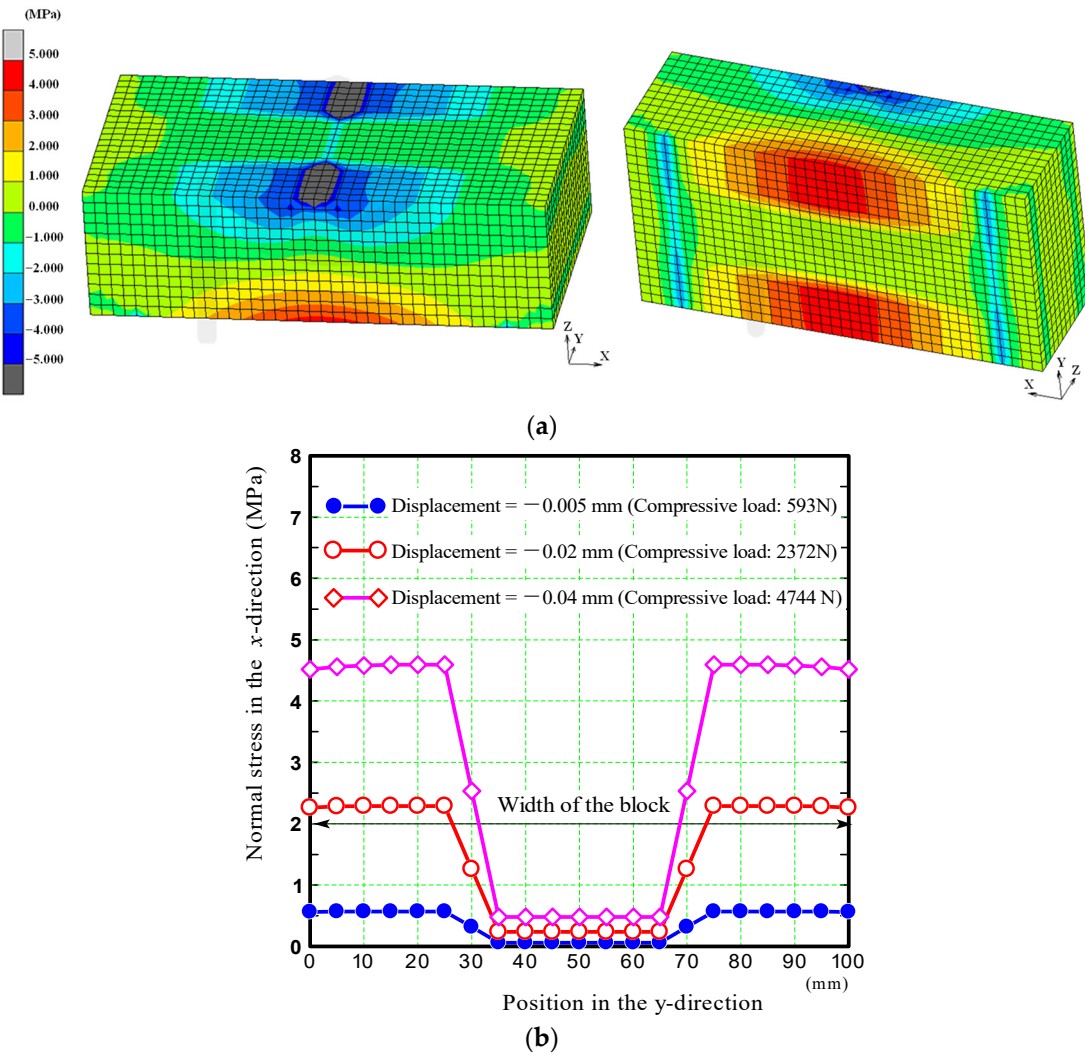

**Figure 15.** (**a**) Contour maps of $\sigma_{xx}$ in the permeable pavement block, and (**b**) the distribution of $\sigma_{xx}$ at the nodes along line A—A at the bottom of the block when the middle of the upper surface of the block was compressed by a constant strain rate.

This loading condition may represent the stress inside the block caused by the three-point bending test more accurately than the previous condition, in which an equally distributed load (45 N/mm over the block width; a total load of 4500 N) was applied to the middle of the upper surface of the block.

Figure 15a indicates that the distribution of stresses caused under this loading condition is similar to the stress distribution illustrated in Figure 11. Figure 15b confirms that the stress in the outer frame (mortar material) increases significantly, whereas the stress in the porous section does not increase substantially when compressive displacement is gradually applied to the middle of the upper surface of the block. This is because the rigidity (elastic modulus) of the outer frame is higher than that of the porous material. Therefore, the failure of the block occurs in the outer frame.

In practice, the middle of a block length is rarely subjected to a constant strain over its entire width. Therefore, we primarily analyzed the scenario where a load was applied over its entire width in the middle of the upper surface of the block.

Fired Bora particles can often lead to higher costs. However, we have identified the advantages of using fired Bora as an aggregate for mortar or concrete by comparing it with aggregates such as river sand, sea sand, and crushed sand. Obtaining river sand in large quantities is difficult owing to the depletion of natural resources. Moreover, the gathering of sea sand is severely restricted from the viewpoint of environmental pollution [34–36], and certain challenges exist in its usage. The water washing of the sand and the separate transportation to the construction site warrant considerable expenses. The transportation of crushed sand over long distances also leads to increased costs. Therefore, procuring locally available aggregate for concrete is desirable to reduce transportation costs. This implies that, in the Southern Kyushu region, Japan, the use of fired Bora as an aggregate for mortar or concrete can be promising.

## 4. Conclusions

In this study, we examined the potential of using the fired volcanic pumice Bora as a substitute for sand in mortar. Furthermore, we investigated the development of a lightweight pavement block with extremely high permeability using fired Bora particles. The results are summarized as follows:

1. The hardness of Bora particles significantly increased when fired at temperatures exceeding 900 °C. This can be attributed to the densification caused by the elimination of fine pores in the structure via sintering and the changes in the crystalline structure.
2. Bora particles fired at 1100 °C can be an efficient substitute for crushed sand used in mortar. Our analysis confirmed that lightweight pavement blocks with extremely high permeabilities can be easily fabricated using these fired Bora particles.
3. The optimal width of the porous material section of the permeable pavement block was estimated to be 30–60 mm based on the bending tests and stress analysis results obtained using FEM.

We could fabricate lightweight pavement blocks with extremely high permeabilities using fired Bora particles. However, when putting the permeable pavement block into practical use, it is desirable that the strength of the porous material used to construct the block is also 3 MPa or higher. Therefore, increasing the strength of the porous material remains a future issue.

The use of Bora in lightweight mortar and lightweight concrete is being considered in the future. Bora, after being mined, can also be used as concrete aggregate, because it is classified into 0–4, 4–7, 7–15, 15–20, 20–25, and 25 mm or larger grain sizes. As such, it can be used in a wide range of concrete applications. Especially for the production of mortar and concrete for various applications in areas where crushed stone is difficult to obtain or on remote islands, fired Bora after particle size adjustment is considered to be an alternative aggregate to crushed stone or crushed sand.

Currently, sand is considered an important infrastructure material for economic development and also serves as a raw material for the manufacturing of daily necessities such as glass products, including liquid crystal panels, and for maintaining biodiversity. However, imports of sand have increased owing to the depletion of sand resources [37]. The use of Bora as a substitute for sand is expected to significantly contribute to curbing the use of sand, which affects all aspects of building a sustainable society.

In recent years, flooding damage in urban environments caused by the occurrence of heavy rainfall in various regions owing to climate change has become a major problem [38]. This study focused on sustainable urban development technology and fabricated a permeable pavement block using fired bora. This block contributes significantly as a countermeasure against flooding.

Therefore, we believe that the results of this study, which focus on the controlled use of sand and countermeasures against heavy rainfall, will make a significant contribution to sustainability.

## 5. Patents

Kinoshita H, Yasui K., Minamimagari M. (2022) Permeable block, Japanese Patent pending; patent application number: 2024-14168 (https://www.j-platpat.inpit.go.jp/c1801/PU/JP-2024-014168/11/ja, accessed on 26 April 2024).

**Author Contributions:** Formal analysis, H.K.; Funding acquisition, K.Y. (Kentaro Yasui); Investigation, Y.S. and K.Y. (Kenshiro Yamamura); Methodology, K.Y. (Kentaro Yasui) and H.K.; Resources, M.M.; Writing—original draft, H.K.; Writing—review and editing, K.Y. (Kentaro Yasui), E.H. and C.M. All authors will be informed about each step of the manuscript process, including submission, revision, revision reminder, etc. via emails from our system or the assigned Assistant Editor. All authors have read and agreed to the published version of the manuscript.

**Funding:** This work was supported by the Takahashi Industrial and Economic Research Foundation.

**Institutional Review Board Statement:** Not applicable.

**Data Availability Statement:** Data are contained within the article.

**Acknowledgments:** This work used research equipment shared in the MEXT Project for promoting the public utilization of advanced research infrastructure (program for supporting the construction of core facilities), grant number JPMXS0440900023.

**Conflicts of Interest:** Author Makoto Minamimagari is employed by the company Nanken Kogyo Co., Ltd. The remaining authors declare that the research was conducted in the absence of any commercial or financial relationships that could be construed as a potential conflict of interest.

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
