# Peer review of "Development of a Lightweight Pavement Block with Extremely High Permeability Using the Volcanic Pumice Bora"

_sustainability, doi:10.3390/su16124888_

Round 1

Reviewer 1 Report

Comments and Suggestions for Authors

It's an interesting study regarding a non-conventional volcanic rock-derived material, which is called Bora for its potential to replace usual fine aggregate particles in PCC. Due to its low density, it can be used to produce lightweight, high permeable concrete blocks. Authors may elaborate on how this potential ultimately benefits what specific applications. Although it was loosely implied that it could be used to produce a lightweight pavement block, it would be much clearer to explicitly describe the ultimate applications of this material. Also, authors should clearly explain the reasoning behind the fire treatment of bora particles. Is it to increase the strength? Is it to decrease the permeability? Is it to optimize strength and permeability? Did you compare the costs of additional fire treatments on bora particles? How do they compare with the cost of sand?

Comments on the Quality of English Language

The manuscript is a very well-written document overall and the authors should be commended for their efforts to make the manuscript clear to understand. 

Reviewer 2 Report

Comments and Suggestions for Authors

1.    Results and Discussion: Figures 5a and 5b, the authors have explained why the hardness of fired bora has a massive increase at temp. increase.

2.    Figure 6. X-ray diffraction (XRD) profiles of Bora particles: why Anorthite is dominant.

3.    Bending Strength of Mortar and Porous Materials Using Fired Bora Particles: Looks all the same elastic modulus and sand/cement has high bending strength, explain.

4.    Bending Strength and Permeability of Permeable Pavement Blocks: the authors have explain the connect points between red, black, and blue lines and how they affect bora bending stress.

5.    FEM, what are the properties of each sample and type of fracture criteria.  

Round 2

Reviewer 2 Report

Comments and Suggestions for Authors

No comments